# Participatory Systems for Personalized Prediction

Hailey James[1], Chirag Nagpal[2], Katherine Heller[3], and Berk Ustun[1]

[1]UC San Diego
[2]CMU
[3]Google

## Abstract

Machine learning models often request personal information from users to assign more accurate predictions across a heterogeneous population. Personalized models are not built to support *informed consent*: users cannot "opt out" of providing personal data, nor understand the effects of doing so. In this work, we introduce a family of personalized prediction models called *participatory systems* that support informed consent. Participatory systems are interactive prediction models that let users opt into reporting additional personal data at prediction time, and inform them about how their data will improve their predictions. We present a model-agnostic approach for supervised learning tasks where personal data is encoded as "group" attributes (e.g., sex, age group, HIV status). Given a pool of user-specified models, our approach can create a variety of participatory systems that differ in their training requirements and opportunities for informed consent. We conduct a comprehensive empirical study of participatory systems in clinical prediction tasks and compare them to common approaches for personalization. Our results show that our approach can produce participatory systems that exhibit large improvements in the privacy, fairness, and performance at the population and group level.

## 1 Introduction

Machine learning models are routinely used to assign predictions to *people* – be it to predict if a patient has a rare disease, the risk that a consumer will default on a loan, or the likelihood that a student will matriculate. Models in such applications are *personalized*, in that they solicit users for their personal data to assign more accurate predictions [1]. In the simplest, most common approach, models are personalized using *group attributes* – i.e., categorical features that encode personal characteristics. For example, models for clinical decision support include group attributes that are *protected* [e.g., `sex` 2], *sensitive* [e.g., `HIV status` 3, 4], *self-reported* [e.g., `hours_of_sleep` 2], or *costly* in that they can only be acquired with time, money, or effort [e.g., `tumor_severity` as detected via CT scan 5 or biopsy 6].

Websites and software applications that solicit personal data from their users are designed to support *informed consent*: users can opt out of providing their personal data, and can see how their data will be used to support their decision [see e.g., GDPR consent banners 7, 8]. In contrast, personalized models do not provide such functionality: users cannot "opt-out" of reporting their personal data to a personalized model, nor tell if a model is using it to improve their predictions. This lack of functionality is alarming as standard techniques for personalization do not improve performance across all users who provide personal data [see 9]. In practice, a personalized model might perform worse or just as well as a *generic model* that did not solicit personal data for users specific personal characteristics. In such cases, personalized models violate the promise of personalization – as users in this group report their personal data without receiving a tailored gain in performance in return.

2022 Trustworthy and Socially Responsible Machine Learning (TSRML 2022) co-located with NeurIPS 2022.

*These effects are prevalent, hard to detect, and hard to fix* [9] – underscoring the need to let users opt out of personalization, and to understand its effects for people like themselves.

In this paper, we propose a new family of machine learning models that operationalize these basic principles of responsible personalization. We call these systems *participatory systems* – i.e., interactive machine learning models that let users report personal data to improve their performance at prediction time. We propose a *model-agnostic* approach for settings where personal information is encoded in group attributes. Our approach starts with a user-specified pool of personalized models, which it arranges within a *reporting tree* – i.e., a tree that represents the sequence of reporting decisions for a user (see Fig. 1). The resulting architecture: (1) lets users opt out of reporting some or all personal data; (2) provides information to support this decision (e.g., expected performance gains; change in prediction); (3) ensures that reporting data leads to an expected gain in performance. In practice, this approach has three major benefits:

*Performance & Fairness*: Our approach builds participatory systems that assign personalized predictions using multiple models. This architecture can use personal data in a way that produces large gains in performance for each reporting group (i.e., users who report a specific subset of personal characteristics). In settings with heterogeneous data distributions, we can avoid performance trade-offs imposed by a single model, and further improve performance by assigning predictions to each group using a personalized model that are specifically built for that group.

*Privacy & Harm Mitigation*: Participatory systems naturally mitigate harm while promoting privacy. Specifically, models that allow users to participate must incentivize participation. In this setup, users who are informed as to the gains of personalization will opt out of report personal data when it unnecessarily reduces performance. In light of this behavior, systems can be "pruned" to avoid soliciting personal data from users who do not experience gains – thereby promoting privacy via data minimization.

*Flexibility*: Our approach can produce three kinds of participatory systems, providing practitioners with multiple options to support informed consent (see Fig. 1). These include: (1) a minimal system, which allows users to opt out of an existing personalized model by training one additional model (i.e., a generic model); (2) a flat system, which allows users to opt into personalization, and further improves personalization using a specific model for each reporting group; (3) a sequential system, which allows users opt into reporting each piece of personal data, and improve personalization using a specific model for each reporting group.

**Contributions**   The main contributions of this work are: 1) We introduce a new kind of prediction model that can support informed consent. 2) We develop a model-agnostic approach to learn a variety of participatory systems that allow users to support informed consent under different training and implementation requirements. 3) We conduct a comprehensive empirical study on real-world datasets in clinical decision support, showing how participation can support consent in a way that improves performance and privacy. 4) We provide a Python package to develop and evaluate participatory personalization systems, available at: https://anonymous.4open.science/r/psc_public-164C/

## 2   Participatory Systems

We consider a supervised learning task where categorical attributes encode personal information. We start with a dataset of $n$ examples $(\boldsymbol{x}_i, y_i, \boldsymbol{g}_i)_{i=1}^n$ where each example contains $d$ features $\boldsymbol{x}_i = [x_{i,1}, \ldots, x_{i,d}] \in \mathbb{R}^d$, a label $y_i \in \mathcal{Y}$, and $k$ group attributes $\boldsymbol{g}_i = [g_{i,1}, \ldots, g_{i,k}] \in \mathcal{G}_1 \times \ldots \times \mathcal{G}_k = \mathcal{G}$ (e.g., $\boldsymbol{g}_i = [\texttt{female}, \texttt{HIV} = +]$). We refer to $\boldsymbol{g}_i$ as the *group membership* of $i$, and to the subset of $\{i \mid \boldsymbol{g}_i = \boldsymbol{g}\}$ examples as *group* $\boldsymbol{g}$. We let $n_{\boldsymbol{g}} := |\{i \mid \boldsymbol{g}_i = \boldsymbol{g}\}|$ denote the number of examples in group $\boldsymbol{g}$, and $m = |\mathcal{G}|$ denote the number of (intersectional) groups.

We use the data to fit a *personalized model* $h_{\boldsymbol{g}} : \mathcal{X} \times \mathcal{G} \to \mathcal{Y}$ by standard empirical risk minimization with a loss function $\ell : \mathcal{Y} \times \mathcal{Y} \to \mathbb{R}_+$. We use $\hat{R}(h)$ and $R(h)$ to denote the *empirical risk* and *true risk* of a $h$. We assume that the personalized model corresponds to the best model trained on the entire training dataset $h_{\boldsymbol{g}} \in \operatorname{argmin} \hat{R}_{h \in \mathcal{H}}(h)$. We evaluate the quality of personalization of a personalized model $h_{\boldsymbol{g}}$ by measuring the gains of personalization for group $\boldsymbol{g}$ in comparison to a *generic model* without group attributes $h_0 \in \operatorname{argmin} \hat{R}_{h \in \mathcal{H}_0}(h)$. For this, we measure the performance of $h_{\boldsymbol{g}}$ for group $\boldsymbol{g}$ when they "misreport" group membership as $\boldsymbol{g}'$. We let $h_{\boldsymbol{g}'} := h(\cdot, \boldsymbol{g}')$ denote a personalized

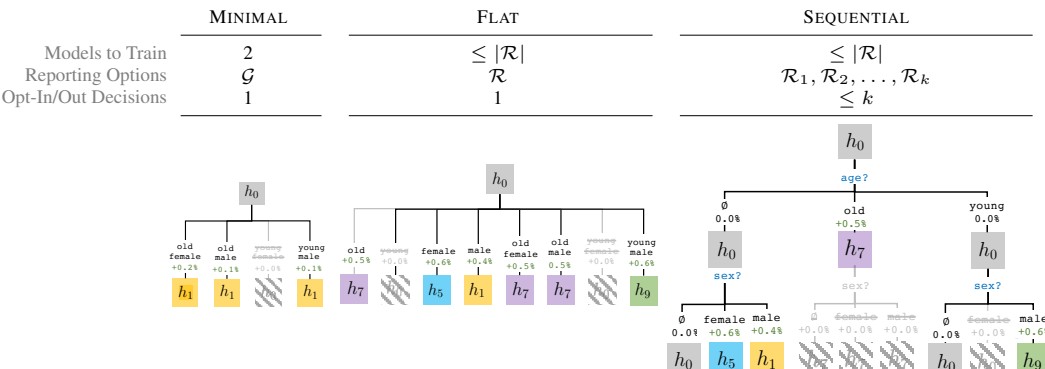

**Figure 1:** Participatory systems for a task where users can opt in/out of reporting $k = 2$ group attributes $\mathcal{R} = \texttt{age} \times \texttt{sex} = [\texttt{male}, \texttt{female}, \emptyset] \times [\texttt{old}, \texttt{young}, \emptyset]$. Each system allows users to opt out of personalization by reporting $\emptyset$, and informs this decision by revealing the gains of personalization (e.g., $+0.2\%$ reduction in error). Each system minimizes data use by removing reporting options that do not lead to gain (e.g., $[\texttt{young}, \texttt{female}]$ is pruned in all systems). We propose three kinds of systems that differ in terms of ease-of-implementation, what users report, and how they report it. The minimal system allows users to opt into a single personalized model, while the flat and sequential models allow for partial reporting and multiple models. In sequential systems, users can can make informed decisions to report each attribute.

model where group membership is fixed to $\boldsymbol{g}'$. Given a personalized model $h_{\boldsymbol{g}}$, we measure its *true risk* and *empirical risk* for group $\boldsymbol{g}$ when they report group membership as $\boldsymbol{g}'$ as:

$$R_{\boldsymbol{g}}(h_{\boldsymbol{g}'}) := \mathbb{E}\left[\ell\left(h(\boldsymbol{x}, \boldsymbol{g}'), y\right) \mid \mathcal{G} = \boldsymbol{g}\right] \qquad \hat{R}_{\boldsymbol{g}}(h_{\boldsymbol{g}'}) := \frac{1}{n_{\boldsymbol{g}}} \sum_{i:\boldsymbol{g}_i = \boldsymbol{g}} \ell\left(h(\boldsymbol{x}_i, \boldsymbol{g}'), y_i\right).$$

Users who provide personal data should expect to receive tailored performance benefits in return. In Definition 1, we formalize this principle in terms of collective preference guarantees.

**Definition 1** (Fair Use, [9]). *A personalized model $h_{\boldsymbol{g}} : \mathcal{X} \times \mathcal{G} \to \mathcal{Y}$ guarantees the fair use of a group attribute $\mathcal{G}$ if it is*

$$\text{'rational' i.e.} \quad R_{\boldsymbol{g}}(h_{\boldsymbol{g}}) \leq R_{\boldsymbol{g}}(h_0) \qquad\qquad \text{for all groups } \boldsymbol{g} \in \mathcal{G}, \tag{1}$$

Condition (1) captures *rationality* for group $\boldsymbol{g}$: a majority of group $\boldsymbol{g}$ prefers a personalized model $h_{\boldsymbol{g}}$ to a generic model $h_0$. The condition is collective, in that performance is measured over individuals in a group, and weak, in that the expected performance gain is non-negative – i.e., no group will be harmed.

This fair use condition enshrines basic expectations of groups in tasks where groups prefer more accurate models. We express these preferences in terms of the *gain* $\Delta_{\boldsymbol{g}}(h, h') := R_{\boldsymbol{g}}(h') - R_{\boldsymbol{g}}(h)$, and make them explicit in Assumption 2.

**Assumption 2** (Rational Preferences). *Given a pair of models $h$ and $h'$, we assume that a group prefers $h$ to $h'$ whenever $\Delta_{\boldsymbol{g}}(h, h') > 0$.*

Assumption 2 holds in applications where individuals prefer to receive correct predictions, such as when estimating disease risk [10, 11, 12] or when receiving content recommendations. This assumption does not hold in settings where individuals may prefer to receive incorrect predictions [see e.g, "polar" clinical prediction tasks in 13]. In insurance pricing, for example, more reliable risk predictions may not be in the best interest of groups whose premiums would increase.

**Participatory Systems** Participatory systems allow individuals to opt in or out of personalization at prediction time. We denote a user's choice to opt out of reporting a group attribute with $\emptyset$. We denote the *reported group membership* for user $i$ as $\boldsymbol{r}_i = [r_{i,1}, \dots, r_{i,k}] \in \mathcal{R} = (\mathcal{G}_1 \cup \emptyset) \times \dots (\mathcal{G}_k \cup \emptyset)$, and the number of reporting groups as $p = |\mathcal{R}|$. Thus, a user with $\boldsymbol{g}_i = [\texttt{female}, \texttt{HIV} = +]$ who only reports sex would have $\boldsymbol{r}_i = [\texttt{female}, \emptyset]$. In Fig. 1, we show participatory systems that differ in terms of what users report and how they report it:

*Minimal systems* let users opt out of a personalized model $h_g$ and receive predictions from its generic counterpart $h_0$. This mechanism allows users to opt out of receiving unnecessarily inaccurate predictions from a personalized model. This setup will improve performance at the group and population level by allowing users to opt into the most accurate predictions from $h_g$ or $h_0$ (since informed rational users would not elect to report information if it does not lead to gain), and may reduce the use of personal data (as we can avoid soliciting information if it does not lead to gain).

*Flat systems* let users report any subset of $2^k$ possible subsets of group attributes. In this setup, users can receive personalized predictions without reporting *all* of personal data. Thus, users can withhold information that they are unwilling or unable to share – e.g., a user with $\boldsymbol{g}_i = [\texttt{age} \geq 50, \texttt{HIV} = \texttt{+}]$ can report $\boldsymbol{r}_i = [\texttt{age} \geq 50, \varnothing]$. Flat systems can further improve performance by assign a distinct personalized model to each reporting group. Thus, users can receive personalized predictions based on a model that is specifically optimized to provide gains for users such as themselves.

*Sequential systems* let users opt into reporting one group attribute at a time. Users make a series of $k$ decisions to report each of $k$ group attributes, and are informed of the gains at each step. Thus, a user with $\boldsymbol{g}_i = [\texttt{age} \geq 50, \texttt{HIV} = \texttt{+}]$ can first report $\texttt{age}$ then decide to report $\texttt{HIV}$. This setup is more tractable in tasks with many group attributes where a flat system may require users to choose between a large number of reporting options (see Fig. 1). In this setting, systems guide users through the sequential decision-making task by revealing: (i) the cumulative performance gain received as a result of all reporting decisions thus far; (ii) the range of additional gains in future steps. Sequential systems are well-suited for settings with optional features – e.g., clinical prediction models where features represent the result of an optional medical procedure [e.g., the Gleason score from a prostate biopsy procedure 5].

**Informing Consent**  Participatory systems can inform consent by providing users with precise information on how their personal data will affect their predictions. This information presented to users should contain, at a minimum, of information that shows: (1) how the additional data will change the expected performance of the system (i.e., the gains of personalization); and (2) how the additional data will change their prediction. In general, the content and format of the information provided should vary based on: (1) the type of system we are building; (2) the performance metric used to measure gains; and (3) the technical expertise of the end-user. In an online medical diagnostic, for example, users would be informed of the expected reduction in error and the probability of error in their diagnosis. This information would ideally be communicated in a way that allows users to account for the uncertainty in estimation [see e.g., 14, 15]. In settings where the diagnostic is soliciting information from patients, one would have to do more work to claim that users truly understand the scope of performance gains [16]. If the patient were assisted by a physician, however, we may be able to present information that is more technical. While our approach can provide flexibility to practitioners in how they compute and present these quantities, we cannot ensure users who consent are truly informed. In practice, implementations of participatory systems should be grounded in best practices from uncertainty quantification, risk communication, and numeracy [17, 18, 19, 20].

## 3  Learning Participatory Systems

### 3.1  Representation

We represent a participatory system as a *reporting tree* where each node is a personalized model assigned to a reporting group (see Fig. 1). Each reporting tree has a generic model at its root, and branches out as users report personal information. Thus, the depth of each tree reflects the number of *reporting decisions* for a user. A flat system, which only allows user to make a 1 opt-in/out decision, corresponds to a $p$-ary tree of depth 1 with $p = |\mathcal{R}|$ leaves, each of which represent the personalized models assigned to each reporting group. A sequential system, which allows users to up to $k$ consecutive opt-in/out decisions, corresponds to a $v$-ary tree with depth $k$ where $k$ is the number of group attributes and $v := \max_t |\mathcal{G}_t|$ is the maximum number of values for any group attribute.

### 3.2  Procedure

We present a model-agnostic procedure to construct participatory systems in Algorithm 1. The input to the system is a pool of candidate models and a validation dataset that is used for assigning and pruning routines. The procedure consists of three routines: (1) enumerate all possible trees (Step 1);

(2) assign a model to each node within the tree (Step 3); (3) prune the trees for data minimization (Step 4). Sequential systems are built using all three routines, while Flat and Minimal systems only require Assignment and Pruning. In what follows, we describe these routines in greater detail.

---

**Algorithm 1** Learning Participatory Systems

Input: $\mathcal{D} = \{(\boldsymbol{x}_i, \boldsymbol{g}_i, y_i)\}_{i=1}^n$          validation dataset
Input: $\mathcal{M} : \{h : \mathcal{X} \times \mathcal{R} \to \mathcal{Y}\}$          pool of candidate models
1: $\mathcal{T} \leftarrow \mathsf{EnumerateTrees}(\mathcal{G})$          *generate all reporting trees*
2: **for** $T \in \mathcal{T}$ **do**          *v-ary trees of models*
3:      $T \leftarrow \mathsf{AssignModels}(T, \mathcal{M})$          *assign models based on*
4:      **repeat**
5:          **for** $r \in \mathsf{leaves}(T)$ **do**          *each tree is an ordering of reporting groups*
6:              $T \leftarrow \mathsf{Prune}(T, \boldsymbol{r})$          *prune models based on*
7:          **end for**
8:      **until** no leaves are pruned
9: **end for**
**Output** $\mathcal{T}$, collection of participatory systems for all reporting groups $\boldsymbol{r} \in \mathcal{R}$

---

**Generating Candidate Models**    We generate a pool of personalized models $h : \mathcal{X} \times \mathcal{R} \to \mathcal{Y}$ that can be assigned to nodes in a reporting tree. This pool should contain a generic model $h_0$ that can be assigned to groups who opt out of reporting all attributes. In practice, we generate the pool by fitting multiple models for each reporting option – i.e., each $2^k$ distinct combination of group attributes that a user could report. The models account for group membership using different personalization techniques (e.g., a one-hot encoding of group attributes, a one-hot encoding of intersectional groups, and variants of these with first degree interaction terms). By default, we include a "decoupled model" for each reporting group that is fit using only data for that group, as such models can perform well on heterogeneous subgroups [9, 21, 22].

**Enumerating Reporting Tree**    We design a custom algorithm for the $\mathsf{EnumerateTrees}$ routine in Step 1 (see Appendix C). This routine is only used for sequential systems since the reporting tree is fixed for minimal and flat systems. Our algorithm enumerates all $k$-ary trees that obey user-specified constraints on ordering and data availability. Thus, one could enforce an ordering constraint to require the trees to solicit lab tests last, allowing patients to avoid lab tests based on other personal characteristics. When used to enumerate the $k$-ary trees for a sequential system, it outputs all possible $v$-vary trees. For a dataset with 3 binary group attributes $\mathcal{G} = \texttt{sex} \times \texttt{age\_group} \times \texttt{blood\_type}$, $\mathcal{T}$ would contain $3^1 \times 2^3 \times 1^9 = 24$ possible 3-ary trees of depth 3. Our routine can scale to datasets with $\leq 8$ group attributes, but does not scale beyond this task. In effect, enumeration $p$-ary trees is intractable as the number of group attributes increases as the number of possible trees is upper bounded by $|\mathcal{T}| \leq \prod_{i=1}^{k} i^{v^{k-i}}$.

**Assigning Models to Reporting Groups**    We assign each reporting group a model using the $\mathsf{AssignModels}$ routine in Step 3. Given a reporting group, we consider all models in the pool that require any subset of personal data that a user could report. Thus, a group who reports $\texttt{age}$ and $\texttt{sex}$ could be assigned a model that requires $\texttt{age}$, $\texttt{sex}$, both, or neither. This implies that we can always assign the generic model to any reporting group, meaning that every system performs at least as well as a generic model in terms of the assignment metric. By default, we assign each reporting group a model from $\mathcal{M}$ that optimizes out-of-sample performance based on a user-specified metric (e.g., 5-CV AUC). This rule can be customized to account for other criteria based on training data (e.g., one can filter $\mathcal{M}$ so that we only consider models that generalize).

**Pruning for Data Minimization**    Algorithm 1 may output trees where it might not make sense for a specific reporting group to report personal data. This could happen in two ways:

1. A tree could assign the same model to a pair of nested reporting groups, which would correspond to a participatory system in which a group who reports personal data receives the same predictions (see e.g., a tree that assigns a generic model to $[\texttt{female}, \emptyset]$ and $[\texttt{female}, \texttt{young}]$ in Fig. 1).

2. A tree could also assign distinct models to a pair of nested groups, which would correspond to a participatory system where a model would report personal only to receive predictions that

are expected to reduce performance (see e.g., Fig. 1, where [female, young] receives better performance from the generic model $h_0$ in the flat system).

In line 4, we Prune each tree to ensure that the corresponding participatory system does not solicit data in such cases. The routine prunes a tree where a leaf that is assigned the same model as its parent by simply checking the assignment (to ensure that the participatory system will not assign the same predictions). In addition, the routine prunes a tree where a leaf that is assigned a model that performs worse than its parent (to ensure that the participatory system only solicits data that can improve predictions). In the latter case, the decision to prune is based on a one-sided hypothesis test that checks if group $g$ prefers the parent model $h$ to the model at the leaf $h'$:

$$H_0 : R_{\boldsymbol{g}}(h) \leq R_{\boldsymbol{g}}(h') \quad \text{vs.} \quad H_A : R_{\boldsymbol{g}}(h) > R_{\boldsymbol{g}}(h') \tag{2}$$

Here, the null hypothesis $H_0$ assumes that a group prefers the parent model $h$ over the model at the leaf $h'$. Thus, we reject $H_0$ when there is enough evidence to suggest that $h'$ performs better for $g$ on a held-out dataset. The testing procedure varies based on the performance metric used to evaluate the gains of personalization. In general, we can apply a bootstrap hypothesis test [23], or choose a more powerful test for common performance metrics [see e.g., the McNemar test for accuracy 24]. In settings where we must test for gains multiple times, we can control for the false discovery rate using a standard Bonferroni correction [25], which is suitable even for non-independent tests.

## 4   Experiments

We present an empirical study of participatory systems on real-world datasets for clinical decision support. Our goals are to compare participatory systems against other kinds of personalized models in terms of performance, data use, and opportunities for informed consent. The software to reproduce the results to our submission can be found here, and we include additional experimental results in Appendix B.

### 4.1   Setup

**Datasets**   We consider six datasets for clinical decision support shown in Table 1 that include group attributes such as sex, age group, or HIV status. We focus on clinical prediction models since they currently require users to report various kinds of personal data that should be optional (e.g., characteristics that are protected, self-reported, sensitive, or costly). We minimally process each dataset to handle missing data, binarize categorical features, and repair class imbalances at the group level. We split each dataset into training sample (60%) used to train models, a validation sample (20%) used to assign and prune models, and a test sample (20%) used to evaluate performance.

**Methods**   We use each dataset to fit 6 kinds of personalized models: (1) 1Hot, a model fit with a one-hot encoding of group attributes; (2) mHot, a model fit with a one-hot encoding of intersectional groups; (3) Impute, a 1Hot model where users can opt out of personalization by imputing their group membership; (4) Minimal, a minimal system composed of 1Hot and its generic counterpart; (5) Flat, a flat system composed of 1Hot, mHot, and their generic counterparts; and (5) Seq: a sequential system composed of 1Hot, mHot, and their generic counterparts. We fit all models – i.e., the personalized models and the components of participatory systems – from a single hypothesis class. We report results for logistic regression, and defer results for random forests to Appendix B.4.[1]

### 4.2   Results

Our results in Table 1 show that participatory systems can use group attributes in ways that improve performance at both the population level and the group level. In particular, participatory systems achieve the best overall and group-level performance on all datasets. In contrast, traditional approaches not only perform worse, but assign unnecessarily inaccurate predictions for specific group on at least 3/6 datasets (see # violations in red). For example, on the saps dataset, we find that mHot

---

[1]In practice, most clinical prediction models are built using logistic regression and a one-hot encoding of group attributes [see e.g., 26, 27, 28]. These simple models are well-suited for this setting since they perform well across multiple performance metrics for clinical decision support (i.e., accuracy, AUC) and generalize in small-sample regimes that arise when working with intersectional groups.

improves Test AUC at a population level but reduces Test AUC for the worst-off group by -0.002, leading to 1 statistically significant fair use violation. This means that at least one group would have been better off with the generic model using a hypothesis test with 10% significance. Our results for Minimal show that simple participatory systems can reap benefits in such cases: when a personalized model assigns unnecessarily inaccurate predictions, a minimal system that allows users to opt out can improve performance and reduce data collection.

**On the Benefits of Complex Participatory Architectures**  Our results highlight some of the benefits of using a flat or sequential system over minimal systems. We find that flat and sequential systems can further improve performance – with gains ranging from small to large (e.g., 0.006 AUC on `lungcancer` vs. 0.085 AUC on `saps`). More complex participatory systems can also solicit less personal data and provide more opportunities for consent. For example, the flat and sequential systems lead to a data reduction of 50% and 25.0% on `cardio_eicu`, meaning that they require 50% to 75% of the data collected by a traditional system. In this dataset, sequential systems provide additional opportunities for consent (e.g., 100% compared to 50.0% for a flat system).

**On the Beneficiaries of Participation**  The ranges of group gain suggest that most groups, and not only those harmed by a static system, benefit from participatory systems. For example, on $5/6$ datasets, both the worse case and best case gains improve for the flat system compared with the static or imputed systems. This translates to better predictions for users across a range of sex, age, and HIV status intersectional groups. These gains are likely a consequence of added capacity provided by the use of multiple models in the flat and sequential systems.

**On the Potential for Data Reduction**  Our results highlight how participatory systems can reap the benefits of personalization without requiring all users to report personal data. In practice, the potential for data reduction varies across datasets and our choice of performance metric.

**On the Pitfalls of Imputation**  Imputation is an alternative way to allow users to opt out of personalization. In theory, imputation could resolve fair use violations when a harmed group is imputed the value of a group that they would have been better off reporting. Here, we impute group membership using mean imputation as an illustrative example. Our results for Impute demonstrate the potential pitfalls of this approach. Although the imputed system does not introduce additional fair use violations and maintains performance across all datasets, we still observe fair use violations on $3/6$ datasets. This suggests that limiting the system to a single model, even with careful imputation, may not achieve the capacity required to mitigate fair use violations.

## 5  Concluding Remarks

This work describes methods for building participatory systems and demonstrates their benefits on real-world clinical prediction tasks. Participatory systems allow users to consent to the use of their personal data and provide them with information that can inform consent. We caution that presenting users with information does not necessarily mean that users will understand the information that is presented to them. Effectively informing users remains a key consideration when implementing participatory systems in practice and an avenue for future work.

One possible limitation of our approach is that it precludes the ability to improve the system over time by collecting additional data in deployment and using it to update the model. This is because participation allows users might opt out of reporting personal data. One solution is to allow individuals to report additional information voluntarily for model improvement.

| | | STATIC | | IMPUTED | PARTICIPATORY | | |
|---|---|---|---|---|---|---|---|
| Dataset | Metrics | 1Hot | mHot | Impute | Minimal | Flat | Seq |
| cardio_eicu $n = 1341, d = 49$ $\mathcal{G} = \{\text{age}, \text{sex}\}$ $m = 4$ Pollard et al. [29] | Overall Performance | 0.858 | 0.857 | 0.858 | 0.858 | **0.923** | **0.923** |
| | Overall Gain | 0.001 | -0.000 | 0.001 | 0.001 | **0.067** | **0.067** |
| | Group Gains | -0.001 – 0.002 | -0.001 – 0.002 | -0.001 – 0.002 | -0.001 – 0.002 | 0.008 – 0.094 | 0.008 – 0.094 |
| | # Violations | 2 | 1 | 3 | 1 | **0** | **0** |
| | Data Reduction | 0.0% | 0.0% | NA% | 0.0% | 50.0% | 25.0% |
| | Opportunity for Consent | 0.0% | 0.0% | NA% | 0.0% | 50.0% | 100.0% |
| cardio_mimic $n = 5289, d = 49$ $\mathcal{G} = \{\text{age}, \text{sex}\}$ $m = 4$ Johnson et al. [30] | Overall Performance | 0.876 | 0.876 | 0.876 | 0.877 | **0.896** | **0.896** |
| | Overall Gain | -0.000 | -0.000 | -0.000 | 0.000 | **0.020** | **0.020** |
| | Group Gains | -0.000 – 0.001 | -0.000 – 0.001 | -0.000 – 0.001 | -0.000 – 0.001 | 0.005 – 0.034 | 0.005 – 0.034 |
| | # Violations | **0** | 2 | **0** | **0** | **0** | **0** |
| | Data Reduction | 0.0% | 0.0% | NA% | 0.0% | 37.5% | 25.0% |
| | Opportunity for Consent | 0.0% | 0.0% | NA% | 0.0% | 40.0% | 100.0% |
| lungcancer $n = 120641, d = 84$ $\mathcal{G} = \{\text{age}, \text{sex}\}$ $m = 6$ NCI [31] | Overall Performance | 0.855 | 0.855 | 0.855 | 0.855 | **0.861** | **0.861** |
| | Overall Gain | 0.001 | 0.001 | 0.001 | 0.001 | **0.007** | **0.007** |
| | Group Gains | -0.000 – 0.000 | -0.000 – 0.000 | -0.000 – 0.000 | -0.000 – 0.000 | 0.001 – 0.012 | 0.001 – 0.012 |
| | # Violations | 2 | 2 | 2 | 1 | **0** | **0** |
| | Data Reduction | 0.0% | 0.0% | NA% | 0.0% | 29.2% | 16.7% |
| | Opportunity for Consent | 0.0% | 0.0% | NA% | 0.0% | 35.3% | 100.0% |
| saps $n = 7797, d = 36$ $\mathcal{G} = \{\text{HIV}, \text{age}\}$ $m = 4$ Allyn et al. [32] | Overall Performance | 0.875 | 0.877 | 0.875 | 0.875 | **0.960** | **0.960** |
| | Overall Gain | 0.010 | 0.011 | 0.010 | 0.009 | **0.095** | **0.095** |
| | Group Gains | -0.000 – 0.015 | -0.002 – 0.019 | -0.000 – 0.015 | 0.000 – 0.015 | 0.035 – 0.139 | 0.026 – 0.139 |
| | # Violations | **0** | 1 | **0** | **0** | **0** | **0** |
| | Data Reduction | 0.0% | 0.0% | NA% | 0.0% | 25.0% | 31.3% |
| | Opportunity for Consent | 0.0% | 0.0% | NA% | 0.0% | 33.3% | 100.0% |
| sleepapnea $n = 1152, d = 26$ $\mathcal{G} = \{\text{age}, \text{sex}\}$ $m = 6$ Ustun et al. [33] | Overall Performance | 0.774 | 0.774 | 0.774 | 0.775 | **0.850** | **0.850** |
| | Overall Gain | -0.002 | -0.002 | -0.002 | -0.001 | **0.074** | **0.074** |
| | Group Gains | -0.002 – 0.002 | -0.002 – 0.003 | -0.002 – 0.002 | -0.002 – 0.002 | 0.004 – 0.115 | 0.004 – 0.115 |
| | # Violations | 2 | 3 | 2 | 1 | **0** | **0** |
| | Data Reduction | 0.0% | 0.0% | NA% | 0.0% | 50.0% | 25.0% |
| | Opportunity for Consent | 0.0% | 0.0% | NA% | 0.0% | 50.0% | 100.0% |
| support $n = 9105, d = 55$ $\mathcal{G} = \{\text{age}, \text{sex}\}$ $m = 6$ Knaus et al. [34] | Overall Performance | 0.707 | 0.706 | 0.707 | 0.706 | **0.712** | **0.712** |
| | Overall Gain | 0.002 | 0.001 | 0.002 | 0.001 | **0.007** | **0.007** |
| | Group Gains | -0.000 – 0.003 | -0.000 – 0.003 | -0.000 – 0.003 | 0.000 – 0.003 | -0.000 – 0.023 | -0.000 – 0.023 |
| | # Violations | **0** | **0** | **0** | **0** | **0** | **0** |
| | Data Reduction | 0.0% | 0.0% | NA% | 0.0% | 66.7% | 33.3% |
| | Opportunity for Consent | 0.0% | 0.0% | NA% | 0.0% | 60.0% | 100.0% |

**Table 1:** Performance and Data Use of personalized models for all datasets. We evaluate the proposed systems in terms of: (i) *Overall Performance*, (ii) *Gain in Personalization* (Overall Population and Group Level), (iii) *# of Fair Use Violations* (detected by a hypothesis test at 10% significance); (iv) *Data Reduction* (average reduction in attributes solicited); and (v) *Opportunity for Consent* (the percentage of solicited attributes for which gains are communicated).

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

# A   Related Work

**Algorithmic Fairness**   Our work is broadly related to a stream of work in algorithmic fairness. We consider a setting where models use group attributes to assign more accurate predictions over a heterogeneous population [see e.g., 35, 36, 37]. Several works discuss the need for models to account for group membership in this setting [e.g., 21, 22, 38, 39, 40, 41, 42, 43], noting that it is otherwise impossible to achieve parity – i.e., to perform equally well for all groups [44, 45, 46, 47, 48, 49, 50]. Parity is an ill-suited goal for personalization because methods to achieve parity can equalize performance by reducing performance for groups who perform well, rather than by improving performance for groups who perform poorly [51, 52, 53, 54]. Participatory systems provide a mechanism to ensure the "fair use" of group attributes [9]. Fair use conditions are collective preference guarantees that incentivize truthful self-reporting for all groups who report personal data – namely, *rationality* and *envy-freeneess* [see e.g.,  22, 43, 55, 56, 57, for other applications].

**Personalization**   We study personalization in models that encode personal information using categorical attributes [see e.g., 26, 38, 58]. Existing work often presumes that personalization will improve performance at a population level. Works that evaluate the gains of personalization [59, 60] often do so at a population level rather than a group level. Modern techniques use personal data to help models perform better by accounting for heterogeneity – e.g., by representing higher-order interaction effects [61, 62, 63] or recursively partitioning data [64, 65, 66, 67].

**Data Privacy & Consent**   Participatory systems support key principles of responsible data use articulated in modern legislation – see e.g., guidelines in the OECD [68], GDPR [7], and California Consumer Privacy Act of 2018  [69]. These include principles like *collection limitation* (i.e., data should be collected with the consent of a data subject, and restricted to only what is necessary) and *purpose specification* (i.e., the purpose of data collection should be clearly specified to users). Model developers currently make difficult decisions regarding what users must report about themselves at prediction time [70]. Our work aims to allow users to make such decisions instead. These goals are aligned with recent work showing that preferences with regards to sharing personal data varies considerably across settings and individuals [71, 72]. In effect, individuals care deeply about their ability to control personal data [73, 74, 75] and that individuals face different costs in collecting, disclosing, or leaking information [76, 77, 78, 79, 80]. Consent should not be assumed even in settings with legal protections [see e.g, 81, who show that underrepresented groups do not consent to report their demographic data in clinical settings].

# B Supporting Material for Experiments

In what follows, we present supporting material for the experiments in Section 4. In Appendix B.2, we include additional information about the datasets. In Section B.1, we include precise definitions of the metrics we report in Table 1. In Appendix B.3, we summarize the performance of component models for the participatory systems . In Appendix B.4, we include tables showing the performance of models and systems built to minimize error (i.e., for decision-making applications), and expected calibration error (i.e., for risk prediction).

## B.1 Evaluation Metrics

**Metrics** We evaluate each model or system in terms of six metrics listed below. We measure performance and gains on a held-out test dataset. We assume that users report all their group attributes when they cannot opt out (e.g., for 1Hot, mHot). When a model or system does allow users to opt out, we assume that users will report their group attributes when it strictly improves performance for their reporting group as per Assumption 2 (i.e., a positive gain in terms of a performance metric on validation data).

*Overall Performance*: The population-level performance of a personalized system/model:. This is computed as a weighted average over all intersectional groups: $\sum_{\boldsymbol{g} \in \mathcal{G}} \frac{1}{n_{\boldsymbol{g}}} R_{\boldsymbol{g}}(h_{\boldsymbol{g}})$.

*Overall Gain*: The population-level gain in performance of a personalized system/model over its generic counterpart: $\sum_{\boldsymbol{g} \in \mathcal{G}} \frac{1}{n_{\boldsymbol{g}}} (R_{\boldsymbol{g}}(h_0) - R_{\boldsymbol{g}}(h_{\boldsymbol{g}}))$.

*Group Gains*: The range of group-level gains of a personalized system/model over its generic counterpart across all groups: $[\min_{g \in \mathcal{G}} R_{\boldsymbol{g}}(h_0) - R_{\boldsymbol{g}}(h_{\boldsymbol{g}}), \max_{g \in \mathcal{G}} R_{\boldsymbol{g}}(h_0) - R_{\boldsymbol{g}}(h_{\boldsymbol{g}})]$.

*# Violations*: The number of reporting groups that receive unnecessarily poor predictions by a personalized system/model. We check this for each reporting group using the one-sided hypothesis test in Eq. (2) with $H_0 : R_{\boldsymbol{g}}(h_{\boldsymbol{g}}) \leq R_{\boldsymbol{g}}(h_0)$. We use a bootstrap hypothesis test with 100 resamples, and count a violation if we reject $H_0$ at 10% significance.

*Data Reduction*: The number of attributes that a system/model will not request from an average user: $\sum_{\boldsymbol{g} \in \mathcal{G}} \frac{1}{n_{\boldsymbol{g}}} A_{\boldsymbol{g}}/A_{h_{\boldsymbol{g}}}$. Here, $A_{h_{\boldsymbol{g}}}$ is the number of attributes requested by a system/model for group $\boldsymbol{g}$, and $A_{\boldsymbol{g}}$ is the maximum number of attributes that $\boldsymbol{g}$ could report.

*Opportunity for Informed Consent*: The number of opt-in decisions that a system/model provides an average user: $\sum_{\boldsymbol{g} \in \mathcal{G}} \frac{1}{n_{\boldsymbol{g}}} I_{\boldsymbol{g}}/A_{\boldsymbol{g}}$. Here, $I_{\boldsymbol{g}}$ is the number of opt-in/out decisions that a system provides for group $\boldsymbol{g}$, and $A_{\boldsymbol{g}}$ is the maximum number of attributes that $\boldsymbol{g}$ could report.

## B.2 Additional Information on Datasets

| Dataset | Reference | Outcome Variable | $n$ | $d$ | $m$ | $\mathcal{G}$ |
|---|---|---|---|---|---|---|
| cardio_eicu | Pollard et al. [29] | patient with cardiogenic shock dies | 1,341 | 49 | 4 | {age, sex} |
| cardio_mimic | Johnson et al. [30] | patient with cardiogenic shock dies | 5,289 | 49 | 4 | {age, sex} |
| lungcancer | NCI [31] | patient dies within 5 years | 120,641 | 84 | 6 | {age, sex} |
| saps | Allyn et al. [32] | ICU mortality | 7,797 | 36 | 4 | {age, HIV} |
| sleepapnea | Ustun et al. [33] | patient has obstructive sleep apnea | 1,152 | 28 | 6 | {age, sex} |
| support | Connors et al. [82] | mortality within 6 months of discharge | 9,105 | 55 | 6 | {age, sex} |

**Table 2:** Datasets used in Section 4. $n$ and $d$ denote the number of examples and features in each dataset, respectively. All datasets are de-identified and available to the public. The cardio_eicu, cardio_mimic, lungcancer datasets require access to public data repositories listed under the references. The saps and sleepapnea datasets must be requested from the authors. The support dataset can be downloaded directly from the URL below.

**cardio_eicu & cardio_mimic** Cardiogenic shock is an acute condition in which the heart cannot provide sufficient blood to the vital organs. We create a cohort of patients who have cardiogenic shock in an intensive care unit (ICU) stay using data from either the Collaborative Research Database V2.0 [29] or MIMIC-III [30]. Here, the outcome variable indicates whether a patient with cardiogenic shock will while in the ICU. The features reflect an exhaustive set of relevant clinical criteria derived

from lab tests and vital signs (e.g. systolic BP, heart rate, hemoglobin count), and reflect measurements obtained up to 24 hours before the onset of cardiogenic shock.

**sleepapnea**    We use the obstructive sleep apnea (OSA) dataset outlined in Ustun et al. [33]. This dataset includes a cohort of 1152 patients where 23% have OSA. We use all available features (e.g. BMI, comobordities, age, and sex) and binarize them, resulting in 26 binary features.

**saps**    The SAPS II score is an ICU risk score used to predict the mortality of critically ill patients in the ICU [10]. The data contains records of 7,797 patients from 137 medical centers in 12 countries. Here, the outcome variable indicates whether a patient dies in the ICU, with 12.8% patient of patients dying. The features reflect comorbidities, vital signs, and lab measurements.

**support**    The support Connors et al. [82] dataset is derived from a study of survival risk score of critically-ill patients who were discharged from the ICU. Here, we have records of 9,105 patients. The outcome variable indicates that a patient has died within six months of discharge. The features cover chronic health conditions(e.g., diabetic status, number of comorbidities), vital signs (e.g., mean blood pressure) and results of lab tests (e.g., white blood cell count). The dataset is publically available for research here: https://biostat.app.vumc.org/wiki/Main/DataSets.

**lungcancer**    We consider a cohort of 120,641 patients who were diagnosed with lung cancer between 2004-2016 and monitored as part of the National Cancer Institute SEER study NCI [31]. Here, the outcome variable indicates if a patient die within five years from any cause, with 16.9% patients died within the first five years from diagnosis. The cohorts only represents patients from Greater California, Georgia, Kentucky, New Jersey and Louisiana, and does not cover patients who were lost to follow up (censored). Age and Sex were considered as group attributes. The features reflect the morphology and histology of the tumor (e.g., size, metastasis, stage, node count and location, number and location of notes) as well as interventions that were administered at the time of diagnosis (e.g., surgery, chemo, radiology).

### B.3 Performance of Component Models for the Participatory Systems

| | | | Training | | | Validation | | | Test | | |
|---|---|---|---|---|---|---|---|---|---|---|---|
| | | | **ERROR** | | | **ERROR** | | | **ERROR** | | |
| Group | Model | Parent | $\Delta_0(h)$ | $\Delta_{pa}(h)$ | $R(h)$ | $\Delta_0(h)$ | $\Delta_{pa}(h)$ | $R(h)$ | $\Delta_0(h)$ | $\Delta_{pa}(h)$ | $R(h)$ |
| - | $h_0$ | $h_0$ | 0.0% | 0.0% | 20.8% | 0.0% | 0.0% | 21.1% | 0.0% | 0.0% | 21.7% |
| negative | $h_6$ | $h_0$ | -0.8% | -0.8% | 18.8% | -0.4% | -0.4% | 19.2% | -0.8% | -0.8% | 19.7% |
| positive | $h_0$ | $h_0$ | 0.0% | 0.0% | 22.0% | 0.0% | 0.0% | 22.6% | 0.0% | 0.0% | 22.8% |
| <30 & positive | $h_3$ | $h_0$ | -12.3% | -12.3% | 0.0% | -13.5% | -13.5% | 0.0% | -14.2% | -14.2% | 0.0% |
| >30 & positive | $h_{26}$ | $h_0$ | -3.1% | -3.1% | 28.6% | -3.1% | -3.1% | 28.9% | -2.7% | -2.7% | 28.6% |

**Table 3:** Group-level performance as measured by error on dataset (saps). $\Delta_0(h)$ represents the change in error compared with the generic classifier (negative is a decrease in error). $\Delta_{pa}(h)$ is the change in error compared with the parent classifier in the reporting tree (see column Parent). $R(h)$ is the error rate for the group. Performance is reported across training, validation and test.

| | | | Training | | | Validation | | | Test | | |
|---|---|---|---|---|---|---|---|---|---|---|---|
| | | | **AUC** | | | **AUC** | | | **AUC** | | |
| Group | Model | Parent | $\Delta_0(h)$ | $\Delta_{pa}(h)$ | $R(h)$ | $\Delta_0(h)$ | $\Delta_{pa}(h)$ | $R(h)$ | $\Delta_0(h)$ | $\Delta_{pa}(h)$ | $R(h)$ |
| - | $h_0$ | $h_0$ | 0.000 | 0.000 | 0.874 | 0.000 | 0.000 | 0.870 | 0.000 | 0.000 | 0.865 |
| negative | $h_9$ | $h_9$ | 0.025 | 0.000 | 0.911 | 0.026 | 0.000 | 0.911 | 0.026 | 0.000 | 0.906 |
| positive | $h_6$ | $h_6$ | 0.011 | 0.000 | 0.881 | 0.011 | 0.000 | 0.876 | 0.011 | 0.000 | 0.871 |
| <30 & negative | $h_{27}$ | $h_9$ | 0.033 | 0.020 | 0.959 | 0.030 | 0.018 | 0.954 | 0.035 | 0.022 | 0.954 |
| <30 & positive | $h_3$ | $h_6$ | 0.082 | 0.075 | 1.000 | 0.092 | 0.086 | 1.000 | 0.101 | 0.093 | 1.000 |
| >30 & positive | $h_{30}$ | $h_6$ | 0.136 | 0.121 | 0.937 | 0.135 | 0.121 | 0.937 | 0.141 | 0.123 | 0.941 |

**Table 4:** Group-level performance as measured by AUC on dataset (saps). $\Delta_0(h)$ represents the change in AUC compared with the generic classifier (positive is an increase in AUC). $\Delta_{pa}(h)$ is the change in AUC compared with the parent classifier in the reporting tree (see column Parent). $R(h)$ is the AUC for the group. Performance is reported across training, validation and test.

## B.4 Additional Experimental Results

| Dataset | Metrics | STATIC | | IMPUTED | PARTICIPATORY | | |
|---|---|---|---|---|---|---|---|
| | | 1Hot | mHot | Impute | Minimal | Flat | Seq |
| cardio_eicu $n = 1341, d = 49$ $\mathcal{G} = \{age, sex\}$ $m = 4$ Pollard et al. [29] | Overall Performance | 22.4% | 21.9% | 23.4% | 21.7% | **16.1%** | **16.1%** |
| | Overall Gain | 0.2% | 0.7% | -0.7% | 0.9% | **6.5%** | **6.5%** |
| | Group Gains | -2.1% − 3.2% | -1.9% − 5.1% | -2.1% − 0.3% | 0.0% − 3.2% | -1.9% − 17.8% | -1.9% − 17.8% |
| | Max Disparity | 5.3% | 7.1% | 2.4% | 3.2% | 19.7% | 19.7% |
| | # Violations | 2 | 2 | 2 | **0** | 1 | 1 |
| | Data Reduction | 0.0% | 0.0% | NA% | 0.0% | 50.0% | 25.0% |
| | Opportunity for Consent | 0.0% | 0.0% | NA% | 0.0% | 50.0% | 100.0% |
| cardio_mimic $n = 5289, d = 49$ $\mathcal{G} = \{age, sex\}$ $m = 4$ Johnson et al. [30] | Overall Performance | 19.5% | 19.3% | 19.1% | 19.2% | **18.1%** | **18.1%** |
| | Overall Gain | -0.3% | -0.1% | 0.1% | 0.0% | **1.1%** | **1.1%** |
| | Group Gains | -0.8% − 0.3% | -0.5% − 0.3% | -0.8% − 0.7% | 0.0% − 0.0% | -0.6% − 3.3% | -0.6% − 3.3% |
| | Max Disparity | 1.1% | 0.8% | 1.5% | 0.0% | 3.9% | 3.9% |
| | # Violations | 2 | 2 | 1 | **0** | 1 | 1 |
| | Data Reduction | 0.0% | 0.0% | NA% | 0.0% | 62.6% | 31.3% |
| | Opportunity for Consent | 0.0% | 0.0% | NA% | 0.0% | 57.2% | 100.0% |
| lungcancer $n = 120641, d = 84$ $\mathcal{G} = \{age, sex\}$ $m = 6$ NCI [31] | Overall Performance | 19.6% | 19.6% | 19.6% | 19.5% | **18.9%** | **18.9%** |
| | Overall Gain | -0.1% | -0.1% | -0.1% | -0.0% | **0.6%** | **0.6%** |
| | Group Gains | -0.4% − 0.1% | -0.3% − 0.1% | -0.4% − 0.0% | -0.1% − 0.0% | 0.3% − 0.9% | 0.4% − 0.9% |
| | Max Disparity | 0.6% | 0.4% | 0.4% | 0.1% | 0.5% | 0.5% |
| | # Violations | 4 | 3 | 4 | 1 | **0** | **0** |
| | Data Reduction | 0.0% | 0.0% | NA% | 0.0% | 25.0% | 41.6% |
| | Opportunity for Consent | 0.0% | 0.0% | NA% | 0.0% | 33.3% | 100.0% |
| saps $n = 7797, d = 36$ $\mathcal{G} = \{HIV, age\}$ $m = 4$ Allyn et al. [32] | Overall Performance | 20.4% | 20.7% | 26.8% | 20.4% | **11.1%** | **11.1%** |
| | Overall Gain | 1.3% | 1.0% | -5.1% | 1.3% | **10.6%** | **10.6%** |
| | Group Gains | 0.0% − 3.6% | 0.0% − 2.7% | -20.8% − 0.7% | 0.0% − 3.6% | 4.3% − 17.2% | 3.9% − 17.2% |
| | Max Disparity | 3.6% | 2.7% | 21.5% | 3.6% | 12.9% | 13.3% |
| | # Violations | **0** | **0** | 2 | **0** | **0** | **0** |
| | Data Reduction | 0.0% | 0.0% | NA% | 0.0% | 37.4% | 31.3% |
| | Opportunity for Consent | 0.0% | 0.0% | NA% | 0.0% | 39.9% | 100.0% |
| sleepapnea $n = 1152, d = 26$ $\mathcal{G} = \{age, sex\}$ $m = 6$ Ustun et al. [33] | Overall Performance | 29.1% | 29.3% | 30.3% | 28.9% | **24.2%** | **24.2%** |
| | Overall Gain | 0.1% | -0.1% | -1.1% | 0.3% | **4.9%** | **4.9%** |
| | Group Gains | -1.1% − 1.2% | -0.8% − 0.4% | -2.7% − 0.4% | 0.0% − 1.2% | 0.0% − 13.8% | 0.0% − 13.8% |
| | Max Disparity | 2.4% | 1.2% | 3.1% | 1.2% | 13.8% | 13.8% |
| | # Violations | 1 | 1 | 3 | **0** | **0** | **0** |
| | Data Reduction | 0.0% | 0.0% | NA% | 0.0% | 58.6% | 29.3% |
| | Opportunity for Consent | 0.0% | 0.0% | NA% | 0.0% | 54.7% | 100.0% |
| support $n = 9105, d = 55$ $\mathcal{G} = \{age, sex\}$ $m = 6$ Knaus et al. [34] | Overall Performance | 35.0% | 35.0% | 35.8% | 35.4% | **34.8%** | **34.8%** |
| | Overall Gain | 0.8% | 0.8% | 0.0% | 0.4% | **1.1%** | **1.1%** |
| | Group Gains | 0.0% − 2.3% | -0.5% − 2.6% | -1.8% − 1.9% | 0.0% − 1.4% | -0.3% − 2.9% | -0.3% − 2.9% |
| | Max Disparity | 2.3% | 3.0% | 3.7% | 1.4% | 3.1% | 3.1% |
| | # Violations | **0** | **0** | 2 | **0** | 1 | **0** |
| | Data Reduction | 0.0% | 0.0% | NA% | 0.0% | 50.0% | 25.0% |
| | Opportunity for Consent | 0.0% | 0.0% | NA% | 0.0% | 50.0% | 100.0% |

**Table 5:** Overview of performance, data use, and consent for all personalized models on all datasets, as measured by *test error*.

| | | STATIC | | IMPUTED | PARTICIPATORY | | |
|---|---|---|---|---|---|---|---|
| Dataset | Metrics | 1Hot | mHot | Impute | Minimal | Flat | Seq |
| cardio_eicu $n = 1341, d = 49$ $\mathcal{G} = \{age, sex\}$ $m = 4$ Pollard et al. [29] | Overall Performance | 0.858 | 0.857 | 0.858 | 0.858 | **0.923** | **0.923** |
| | Overall Gain | 0.001 | -0.000 | 0.001 | 0.001 | **0.067** | **0.067** |
| | Group Gains | $-0.001 - 0.002$ | $-0.001 - 0.002$ | $-0.001 - 0.002$ | $-0.001 - 0.002$ | $0.008 - 0.094$ | $0.008 - 0.094$ |
| | Max Disparity | 0.003 | 0.003 | 0.003 | 0.003 | 0.087 | 0.087 |
| | # Violations | 2 | 1 | 3 | 1 | **0** | **0** |
| | Data Reduction | 0.0% | 0.0% | NA% | 0.0% | 50.0% | 25.0% |
| | Opportunity for Consent | 0.0% | 0.0% | NA% | 0.0% | 50.0% | 100.0% |
| cardio_mimic $n = 5289, d = 49$ $\mathcal{G} = \{age, sex\}$ $m = 4$ Johnson et al. [30] | Overall Performance | 0.876 | 0.876 | 0.876 | 0.877 | **0.896** | **0.896** |
| | Overall Gain | -0.000 | -0.000 | -0.000 | 0.000 | **0.020** | **0.020** |
| | Group Gains | $-0.000 - 0.001$ | $-0.000 - 0.001$ | $-0.000 - 0.001$ | $-0.000 - 0.001$ | $0.005 - 0.034$ | $0.005 - 0.034$ |
| | Max Disparity | 0.001 | 0.001 | 0.001 | 0.001 | 0.028 | 0.028 |
| | # Violations | **0** | 2 | **0** | **0** | **0** | **0** |
| | Data Reduction | 0.0% | 0.0% | NA% | 0.0% | 37.5% | 25.0% |
| | Opportunity for Consent | 0.0% | 0.0% | NA% | 0.0% | 40.0% | 100.0% |
| lungcancer $n = 120641, d = 84$ $\mathcal{G} = \{age, sex\}$ $m = 6$ NCI [31] | Overall Performance | 0.855 | 0.855 | 0.855 | 0.855 | **0.861** | **0.861** |
| | Overall Gain | 0.001 | 0.001 | 0.001 | 0.001 | **0.007** | **0.007** |
| | Group Gains | $-0.000 - 0.000$ | $-0.000 - 0.000$ | $-0.000 - 0.000$ | $-0.000 - 0.000$ | $0.001 - 0.012$ | $0.001 - 0.012$ |
| | Max Disparity | 0.001 | 0.000 | 0.001 | 0.001 | 0.011 | 0.011 |
| | # Violations | 2 | 2 | 2 | 1 | **0** | **0** |
| | Data Reduction | 0.0% | 0.0% | NA% | 0.0% | 29.2% | 16.7% |
| | Opportunity for Consent | 0.0% | 0.0% | NA% | 0.0% | 35.3% | 100.0% |
| saps $n = 7797, d = 36$ $\mathcal{G} = \{HIV, age\}$ $m = 4$ Allyn et al. [32] | Overall Performance | 0.875 | 0.877 | 0.875 | 0.875 | **0.960** | **0.960** |
| | Overall Gain | 0.010 | 0.011 | 0.010 | 0.009 | **0.095** | **0.095** |
| | Group Gains | $-0.000 - 0.015$ | $-0.002 - 0.020$ | $-0.000 - 0.015$ | $0.000 - 0.015$ | $0.035 - 0.139$ | $0.026 - 0.139$ |
| | Max Disparity | 0.015 | 0.020 | 0.015 | 0.015 | 0.105 | 0.114 |
| | # Violations | **0** | 1 | **0** | **0** | **0** | **0** |
| | Data Reduction | 0.0% | 0.0% | NA% | 0.0% | 25.0% | 31.3% |
| | Opportunity for Consent | 0.0% | 0.0% | NA% | 0.0% | 33.3% | 100.0% |
| sleepapnea $n = 1152, d = 26$ $\mathcal{G} = \{age, sex\}$ $m = 6$ Ustun et al. [33] | Overall Performance | 0.774 | 0.774 | 0.774 | 0.775 | **0.850** | **0.850** |
| | Overall Gain | -0.002 | -0.002 | -0.002 | -0.001 | **0.074** | **0.074** |
| | Group Gains | $-0.002 - 0.002$ | $-0.002 - 0.003$ | $-0.002 - 0.002$ | $-0.002 - 0.002$ | $0.004 - 0.115$ | $0.004 - 0.115$ |
| | Max Disparity | 0.004 | 0.005 | 0.004 | 0.003 | 0.111 | 0.111 |
| | # Violations | 2 | 3 | 2 | 1 | **0** | **0** |
| | Data Reduction | 0.0% | 0.0% | NA% | 0.0% | 50.0% | 25.0% |
| | Opportunity for Consent | 0.0% | 0.0% | NA% | 0.0% | 50.0% | 100.0% |
| support $n = 9105, d = 55$ $\mathcal{G} = \{age, sex\}$ $m = 6$ Knaus et al. [34] | Overall Performance | 0.707 | 0.706 | 0.707 | 0.706 | **0.712** | **0.712** |
| | Overall Gain | 0.002 | 0.001 | 0.002 | 0.001 | **0.007** | **0.007** |
| | Group Gains | $-0.000 - 0.003$ | $-0.000 - 0.003$ | $-0.000 - 0.003$ | $0.000 - 0.003$ | $-0.000 - 0.023$ | $-0.000 - 0.023$ |
| | Max Disparity | 0.003 | 0.003 | 0.003 | 0.003 | 0.023 | 0.023 |
| | # Violations | **0** | **0** | **0** | **0** | **0** | **0** |
| | Data Reduction | 0.0% | 0.0% | NA% | 0.0% | 66.7% | 33.3% |
| | Opportunity for Consent | 0.0% | 0.0% | NA% | 0.0% | 60.0% | 100.0% |

**Table 6:** Overview of performance, data use, and consent for all personalized models on all datasets, as measured by *test AUC*.

| | | STATIC | | IMPUTED | PARTICIPATORY | | |
|---|---|---|---|---|---|---|---|
| Dataset | Metrics | 1Hot | mHot | Impute | Minimal | Flat | Seq |
| cardio_eicu | Overall Performance | 0.893 | 0.893 | 0.893 | 0.893 | **0.949** | **0.949** |
| | Overall Gain | 0.003 | 0.002 | 0.003 | 0.003 | **0.059** | **0.059** |
| $n = 1341, d = 49$ | Group Gains | -0.006 – 0.012 | -0.008 – 0.010 | -0.006 – 0.012 | -0.006 – 0.012 | 0.017 – 0.070 | 0.017 – 0.070 |
| $\mathcal{G} = \{age, sex\}$ | Max Disparity | 0.018 | 0.018 | 0.018 | 0.018 | 0.053 | 0.053 |
| $m = 4$ | # Violations | 2 | 2 | 2 | 2 | **0** | **0** |
| Pollard et al. [29] | Data Reduction | 0.0% | 0.0% | NA% | 0.0% | 12.6% | 12.6% |
| | Opportunity for Consent | 0.0% | 0.0% | NA% | 0.0% | 28.6% | 100.0% |
| cardio_mimic | Overall Performance | 0.880 | 0.881 | 0.880 | 0.880 | **0.920** | **0.920** |
| | Overall Gain | -0.000 | 0.001 | -0.000 | 0.000 | **0.039** | **0.039** |
| $n = 5289, d = 49$ | Group Gains | -0.002 – 0.001 | -0.000 – 0.002 | -0.002 – 0.001 | 0.000 – 0.000 | 0.016 – 0.048 | 0.016 – 0.048 |
| $\mathcal{G} = \{age, sex\}$ | Max Disparity | 0.003 | 0.002 | 0.003 | 0.000 | 0.032 | 0.032 |
| $m = 4$ | # Violations | 2 | **0** | 1 | **0** | **0** | **0** |
| Johnson et al. [30] | Data Reduction | 0.0% | 0.0% | NA% | 0.0% | 50.0% | 25.0% |
| | Opportunity for Consent | 0.0% | 0.0% | NA% | 0.0% | 50.0% | 100.0% |
| lungcancer | Overall Performance | 0.849 | 0.849 | 0.849 | 0.848 | **0.856** | **0.856** |
| | Overall Gain | 0.002 | 0.001 | 0.002 | 0.000 | **0.008** | **0.008** |
| $n = 120641, d = 84$ | Group Gains | -0.001 – 0.003 | -0.001 – 0.002 | -0.001 – 0.003 | 0.000 – 0.003 | 0.002 – 0.020 | 0.002 – 0.020 |
| $\mathcal{G} = \{age, sex\}$ | Max Disparity | 0.004 | 0.003 | 0.004 | 0.003 | 0.018 | 0.018 |
| $m = 6$ | # Violations | 1 | 1 | **0** | **0** | **0** | **0** |
| NCI [31] | Data Reduction | 0.0% | 0.0% | NA% | 0.0% | 29.2% | 20.8% |
| | Opportunity for Consent | 0.0% | 0.0% | NA% | 0.0% | 35.3% | 100.0% |
| saps | Overall Performance | 0.921 | 0.922 | 0.921 | 0.922 | **0.966** | **0.966** |
| | Overall Gain | 0.003 | 0.004 | 0.003 | 0.004 | **0.048** | **0.048** |
| $n = 7797, d = 36$ | Group Gains | -0.002 – 0.010 | -0.002 – 0.013 | -0.002 – 0.010 | -0.000 – 0.010 | 0.009 – 0.109 | 0.009 – 0.109 |
| $\mathcal{G} = \{HIV, age\}$ | Max Disparity | 0.012 | 0.015 | 0.012 | 0.011 | 0.100 | 0.100 |
| $m = 4$ | # Violations | 2 | 1 | 2 | 1 | **0** | **0** |
| Allyn et al. [32] | Data Reduction | 0.0% | 0.0% | NA% | 0.0% | 50.0% | 25.0% |
| | Opportunity for Consent | 0.0% | 0.0% | NA% | 0.0% | 50.0% | 100.0% |
| sleepapnea | Overall Performance | 0.825 | 0.824 | 0.825 | 0.824 | **0.944** | **0.944** |
| | Overall Gain | 0.008 | 0.006 | 0.008 | 0.006 | **0.126** | **0.126** |
| $n = 1152, d = 26$ | Group Gains | -0.004 – 0.009 | -0.005 – 0.012 | -0.004 – 0.009 | -0.003 – 0.009 | 0.059 – 0.159 | 0.059 – 0.159 |
| $\mathcal{G} = \{age, sex\}$ | Max Disparity | 0.012 | 0.017 | 0.012 | 0.012 | 0.100 | 0.100 |
| $m = 6$ | # Violations | 2 | 2 | **0** | 1 | **0** | **0** |
| Ustun et al. [33] | Data Reduction | 0.0% | 0.0% | NA% | 0.0% | 41.7% | 25.0% |
| | Opportunity for Consent | 0.0% | 0.0% | NA% | 0.0% | 42.9% | 100.0% |
| support | Overall Performance | 0.695 | 0.698 | 0.695 | 0.695 | **0.722** | **0.722** |
| | Overall Gain | 0.001 | 0.003 | 0.001 | 0.001 | **0.027** | **0.027** |
| $n = 9105, d = 55$ | Group Gains | -0.004 – 0.007 | 0.001 – 0.007 | -0.004 – 0.007 | 0.000 – 0.007 | 0.008 – 0.052 | 0.008 – 0.052 |
| $\mathcal{G} = \{age, sex\}$ | Max Disparity | 0.011 | 0.006 | 0.011 | 0.007 | 0.044 | 0.044 |
| $m = 6$ | # Violations | 2 | **0** | 1 | **0** | **0** | **0** |
| Knaus et al. [34] | Data Reduction | 0.0% | 0.0% | NA% | 0.0% | 41.6% | 25.0% |
| | Opportunity for Consent | 0.0% | 0.0% | NA% | 0.0% | 42.8% | 100.0% |

**Table 7:** Performance and Data Use of personalized models for all datasets, as measured by *test AUC* using random forest component classifiers.

| | | STATIC | | IMPUTED | PARTICIPATORY | | |
|---|---|---|---|---|---|---|---|
| Dataset | Metrics | 1Hot | mHot | Impute | Minimal | Flat | Seq |
| cardio_eicu $n = 1341, d = 49$ $\mathcal{G} = \{age, sex\}$ $m = 4$ Pollard et al. [29] | Overall Performance | 17.9% | 17.5% | 19.2% | 17.7% | **12.9%** | **12.9%** |
| | Overall Gain | 0.9% | 1.2% | -0.4% | 1.1% | **5.9%** | **5.9%** |
| | Group Gains | -0.4% – 3.2% | -0.7% – 2.9% | -1.8% – 0.3% | 0.0% – 3.2% | 2.6% – 8.1% | 2.6% – 8.1% |
| | Max Disparity | 3.5% | 3.6% | 2.1% | 3.2% | 5.5% | 5.5% |
| | # Violations | **0** | 1 | 1 | **0** | **0** | **0** |
| | Data Reduction | 0.0% | 0.0% | NA% | 0.0% | 50.0% | 25.0% |
| | Opportunity for Consent | 0.0% | 0.0% | NA% | 0.0% | 50.0% | 100.0% |
| cardio_mimic $n = 5289, d = 49$ $\mathcal{G} = \{age, sex\}$ $m = 4$ Johnson et al. [30] | Overall Performance | 21.3% | 20.9% | 21.3% | 20.3% | **16.8%** | **16.8%** |
| | Overall Gain | -1.2% | -0.7% | -1.2% | -0.2% | **3.4%** | **3.4%** |
| | Group Gains | -1.9% – -0.6% | -1.1% – -0.3% | -1.8% – -0.7% | -0.7% – 0.0% | 0.5% – 5.0% | 0.5% – 5.0% |
| | Max Disparity | 1.3% | 0.8% | 1.1% | 0.7% | 4.5% | 4.5% |
| | # Violations | 4 | 4 | 4 | 1 | **0** | **0** |
| | Data Reduction | 0.0% | 0.0% | NA% | 0.0% | 50.0% | 25.0% |
| | Opportunity for Consent | 0.0% | 0.0% | NA% | 0.0% | 50.0% | 100.0% |
| lungcancer $n = 120641, d = 84$ $\mathcal{G} = \{age, sex\}$ $m = 6$ NCI [31] | Overall Performance | 20.0% | 20.2% | 20.0% | 20.0% | **19.3%** | **19.3%** |
| | Overall Gain | 0.1% | -0.1% | 0.1% | 0.1% | **0.8%** | **0.8%** |
| | Group Gains | -0.3% – 0.2% | -0.5% – 0.0% | -0.3% – 0.3% | 0.0% – 0.2% | 0.0% – 2.3% | 0.0% – 2.3% |
| | Max Disparity | 0.6% | 0.5% | 0.6% | 0.2% | 2.3% | 2.3% |
| | # Violations | 1 | 4 | 1 | **0** | **0** | **0** |
| | Data Reduction | 0.0% | 0.0% | NA% | 0.0% | 33.3% | 25.0% |
| | Opportunity for Consent | 0.0% | 0.0% | NA% | 0.0% | 37.5% | 100.0% |
| saps $n = 7797, d = 36$ $\mathcal{G} = \{HIV, age\}$ $m = 4$ Allyn et al. [32] | Overall Performance | 14.1% | 15.0% | 17.0% | 13.9% | **9.8%** | **9.8%** |
| | Overall Gain | 0.9% | -0.0% | -1.9% | 1.1% | **5.2%** | **5.2%** |
| | Group Gains | -0.8% – 3.4% | -0.5% – 0.3% | -5.1% – 0.8% | 0.0% – 3.4% | 0.0% – 16.4% | 0.0% – 16.4% |
| | Max Disparity | 4.2% | 0.8% | 5.9% | 3.4% | 16.4% | 16.4% |
| | # Violations | 1 | 1 | 3 | **0** | **0** | **0** |
| | Data Reduction | 0.0% | 0.0% | NA% | 0.0% | 37.3% | 18.6% |
| | Opportunity for Consent | 0.0% | 0.0% | NA% | 0.0% | 36.3% | 100.0% |
| sleepapnea $n = 1152, d = 26$ $\mathcal{G} = \{age, sex\}$ $m = 6$ Ustun et al. [33] | Overall Performance | 26.3% | 26.0% | 26.9% | 26.2% | **12.5%** | **12.5%** |
| | Overall Gain | 1.5% | 1.8% | 0.9% | 1.6% | **15.3%** | **15.3%** |
| | Group Gains | -0.8% – 4.2% | 0.4% – 3.8% | -2.2% – 4.2% | 0.0% – 4.2% | 3.3% – 22.2% | 3.3% – 22.2% |
| | Max Disparity | 5.0% | 3.4% | 6.5% | 4.2% | 18.9% | 18.9% |
| | # Violations | 1 | **0** | 1 | **0** | **0** | **0** |
| | Data Reduction | 0.0% | 0.0% | NA% | 0.0% | 33.5% | 25.0% |
| | Opportunity for Consent | 0.0% | 0.0% | NA% | 0.0% | 37.6% | 100.0% |
| support $n = 9105, d = 55$ $\mathcal{G} = \{age, sex\}$ $m = 6$ Knaus et al. [34] | Overall Performance | 36.0% | 35.9% | 35.9% | 35.8% | **35.6%** | **35.6%** |
| | Overall Gain | -0.3% | -0.2% | -0.2% | -0.0% | **0.1%** | **0.1%** |
| | Group Gains | -0.9% – 0.2% | -1.2% – 1.3% | -1.0% – 0.9% | -0.8% – 0.2% | -1.6% – 1.4% | -1.6% – 1.1% |
| | Max Disparity | 1.2% | 2.5% | 1.9% | 1.0% | 3.1% | 2.7% |
| | # Violations | 3 | 3 | 4 | 1 | 1 | 1 |
| | Data Reduction | 0.0% | 0.0% | NA% | 0.0% | 33.4% | 33.3% |
| | Opportunity for Consent | 0.0% | 0.0% | NA% | 0.0% | 37.5% | 100.0% |

**Table 8:** Performance and Data Use of personalized models for all datasets, as measured by *test error* using random forest component classifiers.

# C   Supporting Material for Section 3

In what follows, we provide details on the routine used for the EnumerateTrees procedure in Algorithm 1. We summarize the routine in Algorithm 2, and discuss it below.     The input to Algorithm 2 is an

---

**Algorithm 2** Routine to Enumerate All Possible Reporting Trees for Reporting Options $\mathcal{R}$

---

1: **procedure** ENUMERATETREES($\mathcal{R}$)
2:     **if** $\dim(\mathcal{R}) = 1$ **return** $[T_\mathcal{R}]$                *base case: we are left with only a single attribute on which to branch*
3:     AllTrees $\leftarrow$ [ ]
4:     **for** $\mathcal{A}$ in $\mathcal{R}$ **do**                                            *Each attribute in list of attributes $\mathcal{R}$*
5:         $T_\mathcal{A} \leftarrow$ reporting tree with $n_\mathcal{A} := |\mathcal{A}|$ leaves
6:         $\mathcal{U} \leftarrow$ unsolicited attributes $\mathcal{R} \setminus \mathcal{A}$
7:         AllSubtrees $\leftarrow$ ENUMERATETREES($\mathcal{U}$)                *All subtrees using all attributes except $\mathcal{A}$*
8:         **for** $\mathcal{P}$ in ALLPERMUTATIONS(AllSubTrees, $n_\mathcal{A}$) **do**:                *Each permutation of $n_\mathcal{A}$ subtrees*
9:             $T_{a,\mathcal{P}} \leftarrow T_a$.copy()
10:            $T_{a,\mathcal{P}} \leftarrow T_{a,\mathcal{P}}$.assign_to_leaves($\mathcal{P}$)         *assign_to_leaves extends the tree by assigning subtrees to each leaf*
11:            AllTrees $\leftarrow$ AllTrees $\cup\, T_{a,s}$
12:        **end for**
13:    **end for**
14:    **return** AllTrees, set of all distinct reporting trees for reporting options $\mathcal{R}$
15: **end procedure**

---

ordered collection of reporting options $\mathcal{R}$. The algorithm uses the reporting options to construct the set of all possible reporting trees, each of which branches on all of the attributes in $\mathcal{R}$. At a high level, Algorithm 2 recurses through the attributes one at a time, building trees that begin with each attribute sequentially. Enumerating all possible trees ensures we can recover the best tree given the selection criteria and allows for flexible post-hoc selection criteria (e.g., let a developer choose among the top $k$ trees). In settings constrained by computational resources, we can impose additional stopping criteria and modify the ordering such that we enumerate more plausible trees first or exclusively (e.g., by changing the ordering of $\mathcal{R}$ or imposing constraints in ALLPERMUTATIONS).

