# OpenReview forum: "Participatory Systems for Personalized Prediction"
_NeurIPS.cc/2022/Workshop/TSRML — TSRML2022_

### Official Review · Reviewer_rJZs · 2022-10-19

**Overall Rating:** 7

**Summary:**

* This work introduces prediction algorithms which promote rational disclosure of sensitive information (“participatory systems”).
* Each user $i$ is associated with a feature vector $x_i\in\mathbb{R}^d$ and a $k$-dimensional group membership vector $\mathbf{g}_i$. Group membership is assumed to be a sensitive attribute.
* At training time, all group attributes are assumed to be available. At inference time, each user decides whether to reveal their group attributes, or use the predictions of a baseline model which does not rely on the hidden attributes.
* Each group $\mathbf{g}$ may be associated with a classifier $h_\mathbf{g}$, and disclosure of group attributes is assumed to be the rational choice if $R(h_\mathbf{g}) \le R(h_0)$ for all existing $h_\mathbf{g}$, where $R(h)$ is the expected risk of model $h$.
* Authors present an algorithm which learns participatory prediction algorithms. The algorithm relies on combinatorial enumeration of all disclosure patterns (“reporting trees”).
* In the experiments section, authors empirically evaluate the performance of the algorithm using semi-synthetic simulations based on 6 healthcare datasets. Simulation results suggest significant performance gains.


**Strengths:**

* Problem is well-motivated, and the approach is interesting.
* Empirical evaluation shows statistically-significant gains.
* Documented code is provided.


**Weaknesses:**

* The core ‘rationality’ assumption of the model seems to have some gaps:
  * For example, it is reasonable to assume that users are already aware of their feature vector $x_i$ when making a disclosure decision.
  * However, in Definition 1 it is assumed that disclosure decisions are made according to the expected risk $R(h)$, $R(h_g)$. This implies that users don’t use their knowledge of $x$ when making ‘rational’ disclosure decisions.
  * Additionally, in real-world cases there are likely to be values of $x$ for which $h_0(x)$ provides a correct prediction, while $h_g(x)$ consistently makes mistakes. Agents being aware of their value of $x$ may strategic choose not to reveal their attributes in such cases - reducing adherence, and thus possibly undermining the performance participatory systems ignoring such behavior.
* Method presented is not computationally scalable with respect to the number of disclosable attributes (e.g line 160: “Routine can scale to datasets with $\le 8$ group attributes”). Furthermore, the exponential nature of the algorithm means that the training set size for each group $\mathbf{g}$ decreases exponentially with the number group attributes. This may make training statistically infeasible for even a modest number of group attributes.


**Overall Recommendation:**

Despite the gaps in the behavioral assumptions and the limited scalability of the presented algorithms, the overall setting is intriguing, and I believe the work can serve as a starting point for further exploration.


**Review Confidence:**

3: The reviewer is fairly confident that the evaluation is correct

---

### Official Review · Reviewer_j1Gn · 2022-10-21
**Well written, interesting problem**

**Overall Rating:** 7

**Summary:**

This paper proposes a participatory system that personalises to each specific group and allows individuals to consent to the use of their personal data in an informed way. The model has been validated in clinical validation tasks and extensive experiments have been conducted using six different datasets.

**Strengths:**

The paper investigates an interesting question of personalised model with more flexibility for users to opt in/out their data. It is also relatively well written, and the paper structure is clear.

**Weaknesses:**

•	The number of models is dependent on the number of the group attributes. Currently the authors have only validated two group attributes of age and gender. How would the authors view this system to scale with more attributes, in terms of the accuracy and the computational cost?

•	It is still a bit unclear how the model handles the situation when the user opts out part of the historical data after the model has already been trained with this kind of data (if I understand the model mechanism correctly)? Also, some of the concepts in the proposed model is similar as in reinforcement learning, by collecting the feedbacks/system gains to improve the model. How does the authors view the relationship between this two? This could be slightly clarified.

•	In page 4, in minimal system subsection, typo with ‘It is is bound to’.


**Overall Recommendation:**

This paper investigates an interesting problem and good results also support the effectiveness of the methods. The evaluation is relatively well analysed.

**Review Confidence:**

3: The reviewer is fairly confident that the evaluation is correct

---

### Decision · Program_Chairs · 2022-10-23

Accept